# Rapid Detection of *Staphylococcus aureus* in Milk and Pork via Immunomagnetic Separation and Recombinase Polymerase Amplification

Runan Cheng,[a] Lei Li,[a] Sihui Zhen,[a] Honglei Liu,[a] Zhouhui Wu,[a] Yu Wang,[a] Zhen Wang[a]

aBeijing Key Laboratory of Traditional Chinese Veterinary Medicine, Animal Science and Technology College, Beijing University of Agriculture, Beijing, China

**ABSTRACT** Separation processes using immunomagnetic beads (IMBs) are advantageous for the rapid detection of *Staphylococcus aureus* (*S. aureus*). Herein, a novel method, based on immunomagnetic separation using IMBs and recombinase polymerase amplification (RPA), was employed to detect *S. aureus* strains in milk and pork. IMBs were formed by the carbon diimide method using rabbit anti-*S. aureus* polyclonal antibodies and superparamagnetic carboxyl-$Fe_3O_4$ MBs. The average capture efficiency for 2.5 to $2.5 \times 10^5$ (CFU)/mL gradient dilution of *S. aureus* with 6 mg of IMBs within 60 min were a range of 62.74 to 92.75%. The detection sensitivity of the IMBs-RPA method in artificially contaminated samples was $2.5 \times 10^1$ CFU/mL. The entire detection process was completed within 2.5 h, including bacteria capture, DNA extraction, amplification, and electrophoresis. Among 20 actual samples, one case of raw milk sample and two cases of pork samples were tested positive using the established IMBs-RPA method, which were verified by the standard *S. aureus* inspection procedure. Therefore, the novel method shows potential for food safety supervision owing to its short detection time, higher sensitivity, and high specificity.

**IMPORTANCE** Our study established IMBs-RPA method, which simplified the steps of bacteria separation, shortened the detection time, and realized the convenient detection of *S. aureus* in milk and pork samples. IMBs-RPA method was also suitable for the detection of other pathogens, providing a new method for food safety monitoring and a favorable basis for rapid and early diagnosis of diseases.

**KEYWORDS** carboxyl-$Fe_3O_4$, polyclonal antibodies, immunomagnetic beads, recombinase polymerase amplification, *Staphylococcus aureus*

**S**taphylococcus aureus is a Gram-positive opportunistic zoonotic pathogen that is commonly found airborne in outdoor environments, in the nasal cavity and skin of humans, and in some cases in food. *S. aureus* poses threats to human and animal health because it can cause a variety of diseases, such as purulent infection (1), foodborne poisoning (2), severe community-acquired pneumonia (3), endocarditis (4), and sepsis (5). It is also one of the main bacterial pathogens that cause mastitis in cattle and sheep (6). Cows diagnosed with mastitis required more artificial inseminations and days open, and experienced pregnancy loss more often than healthy cows (7). Furthermore, methicillin-resistant *S. aureus* (MRSA) has also been detected in milk, dairy products, and meat in recent years (8–11). Emerging cross-host transmission of *S. aureus* has been associated with food chains and livestock occupations (12). *S. aureus* infections cause enormous economic losses to the livestock industry. Therefore, the monitoring and detection of *S. aureus* are critical for the prevention and control of these infections.

Since the first report on the synthesis of Fe nanoparticles by Gleiter et al. (13), magnetic nanoparticles (MNPs) based on $\gamma$-$Fe_2O_3$ and $Fe_3O_4$ have received considerable research attention due to their superparamagnetism, high saturation magnetic strength,

Address correspondence to Zhen Wang, wangzhen3355@163.com.

The authors declare no conflict of interest.

excellent magnetic response, excellent biocompatibility, and easy surface functionalization. These properties make $\gamma$-$Fe_2O_3$ and $Fe_3O_4$ magnetic beads (MBs) suitable for biomedical, data storage, spintronics, catalytic, neural stimulations, and gyroscopic sensing applications (14). Among the different methods employed to synthesize MBs, solvothermal method yields $Fe_3O_4$ MBs with controllable sizes, narrow size distributions, hydrophilic surfaces, and strong magnetic responses (15–17). Recent developments in biomedicine have been focused on magnetic drug targeting (18), magnetic hyperthermia (19), magnetic resonance imaging (20), and magnetic diagnostics (21). Fe oxide-based MNPs are also suitable for the pretreatment of biological fluid samples containing low-abundance targets due to their desirable properties. MBs labeled with immune ligands, such as antibodies and aptamers, can capture targets like cells, proteins, DNA, RNA, bacteria, and viruses through antigen–antibody immune reactions and then directly separate the specific targets from the complex matrices under an external magnetic field to enrich the targets and simplify the assays. This procedure can be followed by quantitative PCR (qPCR) (22), enzyme-linked colorimetry (23), real-time recombinase polymerase amplification (qRPA) (24), chemiluminescence (25), immunochromatography, and chromatography to improve the convenience, sensitivity, and accuracy of detection. For example, Ulusoy et al. (26) combined multiwalled carbon nanotubes with $Fe_3O_4$ MBs to extract trace vitamin $B_{12}$ from food samples. In addition, MBs can be used to develop automated equipment to improve detection efficiency (23).

The conventional detection method of *S. aureus* was based on microbial culturing, followed by biochemical identification and antimicrobial susceptibility testing. Molecular biological assays for *S. aureus* like polymerase chain reaction (PCR) (27), real-time PCR (28) and Multiplex real-time PCR (29) are commonly used. In addition, high-performance liquid chromatography (HPLC) (30) and enzyme-linked immunosorbent assay (ELISA) (31) are also used. These methods are greatly affected by the sample matrix, etc., the fat, proteins and other organic matters presented in actual samples, often involve complex operating procedures, and require high-cost equipment. They cannot be used for field or on-site detection. Therefore, rapid, accurate, and sensitive *S. aureus* detection methods must be developed.

Recombinase polymerase amplification (RPA) is a new DNA amplification technology proposed by Piepenburg et al. in 2006. Within a short span of time since then, RPA has been widely used in various applications, such as in the diagnosis of human and animal infectious diseases (32–35), determination of food composition (36), and detection of foodborne pathogens (37). In this method, DNA can be amplified within 10 to 20 min at 37 to 42°C (38). As such, this technology can also be used for the rapid detection of pathogenic bacteria. This way, high sensitivity and specificity can be achieved, while eliminating the complicated process of initial DNA amplification, which includes denaturation, annealing, and extension. Thereafter, the amplification products can be observed by agarose gel electrophoresis, real-time fluorescence, and lateral flow (LF) strips (39).

In this study, a novel, rapid, highly sensitive, and specific alternative method, based on the combination of immunomagnetic separation and RPA, was developed for the on-site detection of food pathogens. Immunomagnetic beads (IMBs) were prepared and then used to rapidly separate and enrich *S. aureus* from milk and pork samples (Fig. 1). The results of the RPA were verified using PCR and standard *S. aureus* inspection procedure (National Standard of the People's Republic of China, GB 4789.10-2016 *Staphylococcus aureus* Inspection Standard).

## RESULTS

**Physical and chemical properties of the $Fe_3O_4$ MBs.** Fig. 2A shows the SEM image of the synthesized $Fe_3O_4$ MBs. The $Fe_3O_4$ MBs were composed of highly aggregated smaller grains. On the other hand, the TEM image shown in Fig. 2B reveals that spherical nanoparticles with an average particle size of 200 nm were successfully prepared.

Fig. 2C shows the magnetization of the $Fe_3O_4$ MBs as a function of the applied magnetic field measured using a vibration sample magnetometer. The saturation magnetization

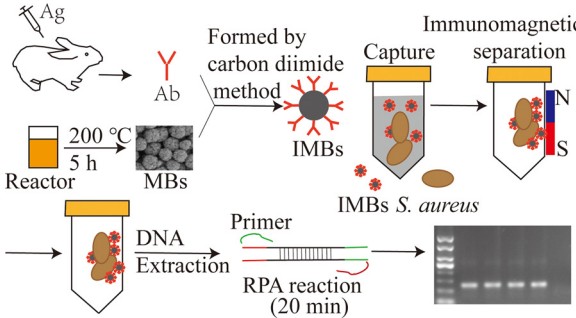

**FIG 1** Design of RPA rapid detection method for *S. aureus* based on IMBs enrichment. The SEM image of the carboxyl-Fe$_3$O$_4$ MBs and the agarose gel electrophoresis image in Fig. 1 were the results of this study.

and coercivity of the Fe$_3$O$_4$ MBs were 74.37 emu/g and 40 Oe, respectively. Therefore, the synthesized Fe$_3$O$_4$ MBs exhibited a strong magnetic response and superparamagnetism.

Fig. 2D shows the FTIR spectrum of the Fe$_3$O$_4$ MBs. The characteristic absorption peak at 583 cm$^{-1}$ can be attributed to the Fe-O bond. Peaks corresponding to the asymmetric ($\nu$[COO-] $_{asym}$) and symmetric ($\nu$[COO-] $_{sym}$) stretching vibrations of the carbonyl group were also observed at 1,536 and 1,411 cm$^{-1}$, respectively. In addition, the characteristic absorption bands observed at 3,436, 2961, and 1,042 cm$^{-1}$ can be attributed to the stretching vibrations of the O-H, C-H, and C-O bonds, respectively. Lastly,

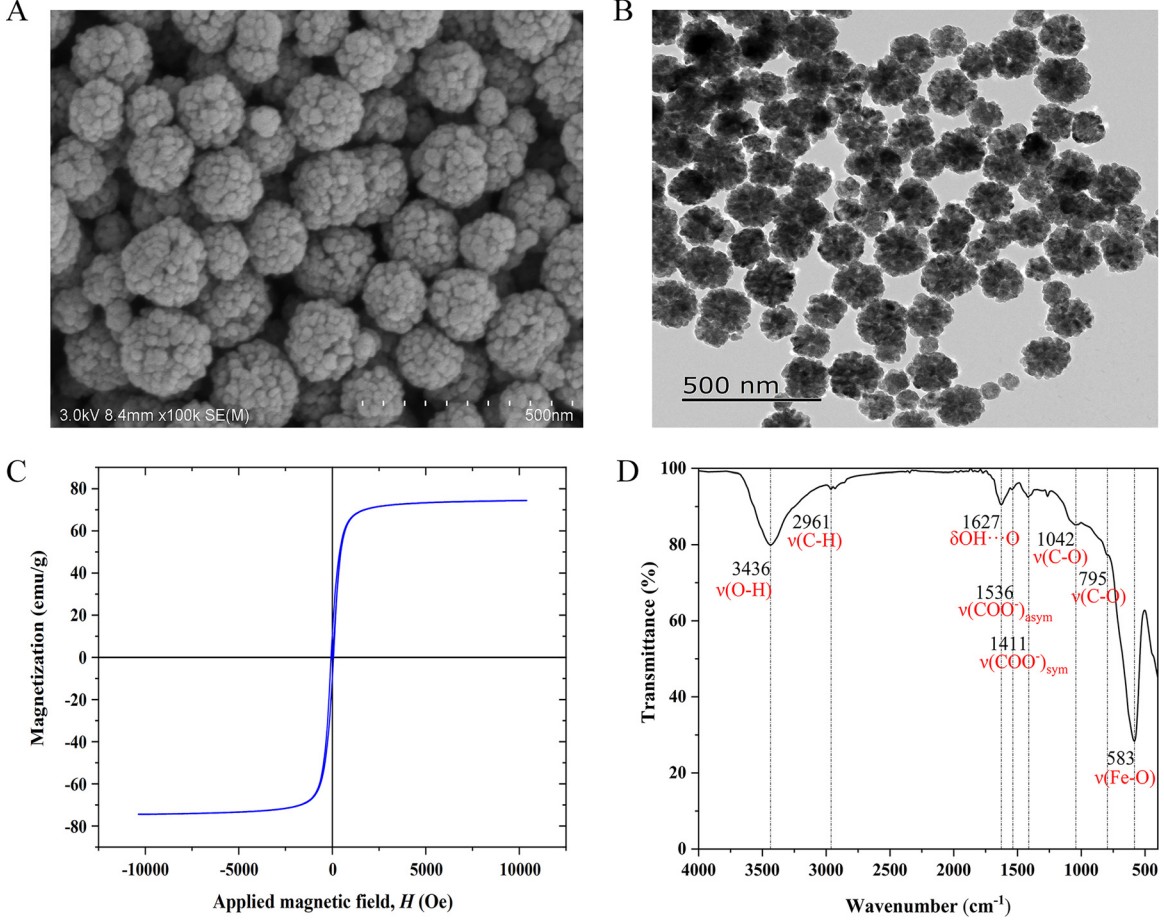

**FIG 2** (A) SEM and (B) TEM images, (C) magnetization response measured using VSM, and (D) FTIR spectrum of the carboxyl-Fe$_3$O$_4$ MBs.

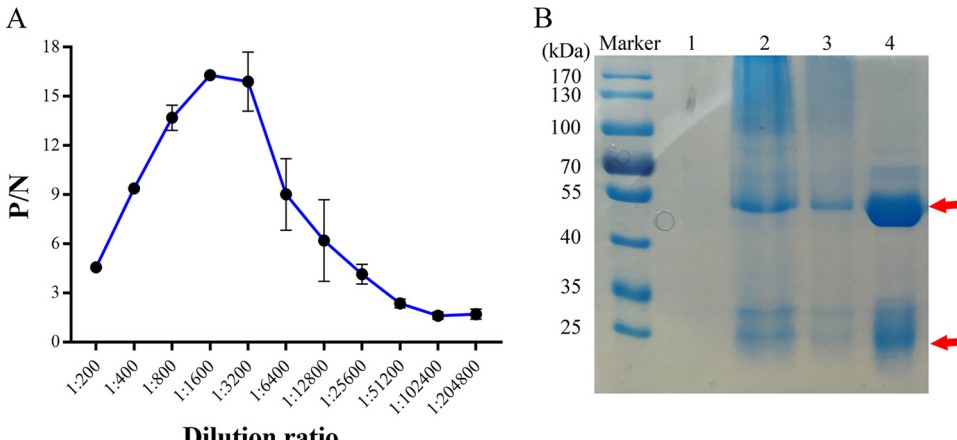

**FIG 3** (A) ELISA titer detection of the *S. aureus* polyclonal antibodies. (B) Detection of the purified antibody, MBs and IMBs through SDS–PAGE. On gel: lane marker (protein marker), lane 1 (MBs without polyclonal antibodies), lane 2 and 3 (IMBs coupled with Polyclonal antibodies), and lane 4 (purified antibody). Red arrows indicate the heavy and light chains of immunoglobulin (IgG).

the absorption band at 1627 cm$^{-1}$ corresponds to the out-of-plane bending vibration of the –OH group ($\delta$ OH···O out-of-plane).

**Performance of polyclonal antibodies and conjugating with Fe$_3$O$_4$ MBs.** The ELISA antibody titer of the *S. aureus* polyclonal antiserum was 1:51,200 (Fig. 3A). After the purification through the protein A method, the concentration of the purified antibody was 11.2 mg/mL. Fig. 3B (lane 4) shows the purity of the antibody obtained through SDS-PAGE. Obvious bands were observed around 50 and 20 kDa, which correspond to the heavy and light chains of IgG, respectively. These results reveal the high purity of the antibody. After the activated MBs were coupled with the antibody in 0.1 M MES buffer (containing 100 mg/mL EDC and 100 mg/mL NHS), bands around 50 and 20 kDa were observed in lanes 2 and 3 (Fig. 3B), indicating that the antibody was successfully adsorbed on the surface of the MBs.

**Capture efficiency of the IMBs to *S. aureus* cultures in PBS.** Gradient-diluted *S. aureus* culture was used to investigate the sensitivity of the prepared IMBs. The highest capture efficiency of the IMBs were 92.75% at $2.5 \times 10^1$ CFU/mL. When there were only 2.5 CFU/mL, the capture efficiency can still reach more than 60% (Fig. 4A). In addition, *Salmonella enterica* serovar Typhimurium, *E. coli*, and *L. monocytogenes* strains were used to investigate the specificity of the IMBs at a concentration of 10$^4$ CFU/mL. The capture efficiency of the IMBs for *S.* Typhimurium, *L. monocytogenes*, and *E. coli* strains were 0.04%, 0.75%, and 0.92%, respectively, which were significantly lower than that for *S. aureus* ($P < 0.001$, Fig. 4B). Interestingly, we also made a specificity test of the IMBs for a mixed sample, the proportions of *S. aureus*, *E. coli*, *S.* Typhimurium, and *L. monocytogenes* in the mixed sample were all 25%. According to the plate count results, the number of *S. aureus* captured by IMBs accounted for 21% of the total number of bacteria in the mixed sample. The result indicated that the IMBs also can effectively capture *S. aureus* in the presence of other bacteria (Fig. 4B). Moreover, the TEM image shown in Fig. 4D reveals that the IMBs were adsorbed onto the surface of *S. aureus*. These results indicate that the prepared IMBs have good capture feature toward *S. aureus*.

**Application of the IMBs-RPA method to spiked milk and pork samples.** The amplicons obtained using the *nuc*-2.1 primer showed the brightest bands and exhibited highest amplification efficiency (Fig. 5A). The product size was 130 bp. The detection line of PCR for *S. aureus* was $2.5 \times 10^3$ CFU/mL (Fig. 5B), while RPA was $2.5 \times 10^1$ CFU/mL (Fig. 5C). Moreover, no bands were observed in the specificity test of the RPA using *S.* Typhimurium, *L. monocytogenes*, and *E. coli* (Fig. 5D). Therefore, the RPA method has a high sensitivity and specificity toward the detection of *S. aureus*. The detection lines of the IMBs-RPA and IMBs-PCR assays for *S. aureus* in spiked milk and

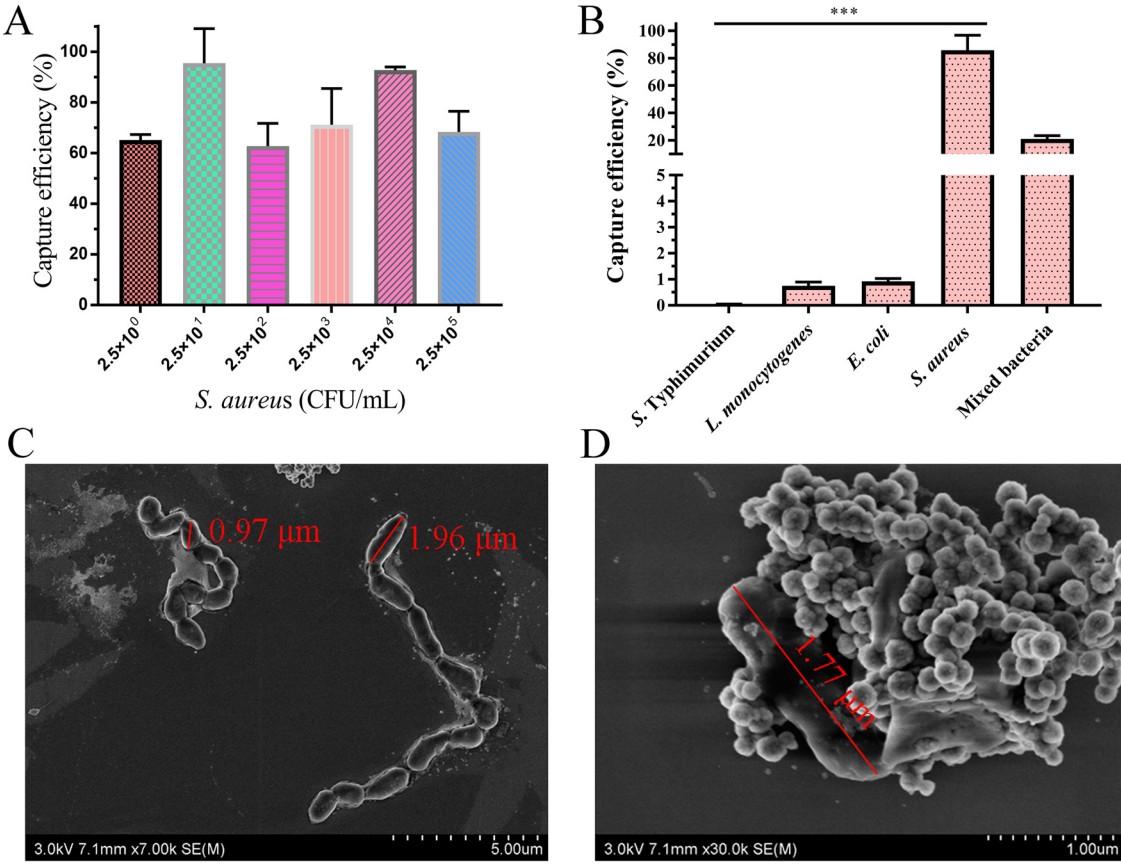

**FIG 4** (A) Capture efficiency of the prepared IMBs (5 mg) to *S. aureus* cultures with different concentrations. (B) Specificity of IMBs for selected single bacteria and mixed bacteria with a concentration of $10^4$ CFU/mL. *** indicates significant difference ($P < 0.001$). (C) TEM image of the *S. aureus* strain and (D) IMBs bind to *S. aureus*.

pork samples were both $2.5 \times 10^1$ CFU/mL (Fig. 5E to H). These results suggest that the sensitivities of the IMBs-RPA and IMBs-PCR methods were approximately 100 times higher than that of the PCR assay alone.

**Detection of *S. aureus* in actual samples using the IMBs-RPA method.** The presence of *S. aureus* in 20 raw milk and 20 chilled pork samples were monitored through the IMBs-RPA method. Among these samples, one milk sample was tested positive for *S. aureus* (Fig. 6A). On the other hand, *S. aureus* was detected in two chilled pork samples (Fig. 6B). These results agree well to those obtained through PCR (Fig. S1) and standard *S. aureus* inspection procedure. According to standard procedure, one case of raw milk sample and two cases of pork samples among 20 actual samples showed black *S. aureus* single colonies on Baird-Parker plate and hemolysis on blood plate, and the Gram staining and coagulase test was positive, which was consistent with the detection results of IMBs-RPA method established in this study. Furthermore, the detection time of the IMBs-RPA was greatly shortened than the standard procedure.

## DISCUSSION

*S. aureus* not only is an important pathogen of food safety problems, but also the main pathogenic bacteria causing mastitis of cattle and sheep. It endangers human and animal health, and brings large economic loss. Therefore, the detection and prevention of *S. aureus* is particularly important. Improving the detection capacity facilitates the accurate and timely identification of food safety issues, which in turn avoids physical and economic losses. However, the sensitivity, specificity, and timeliness of detection methods are greatly affected by the pretreatment of the samples to be tested. Though the traditional culture method has excellent sensitivity and specificity,

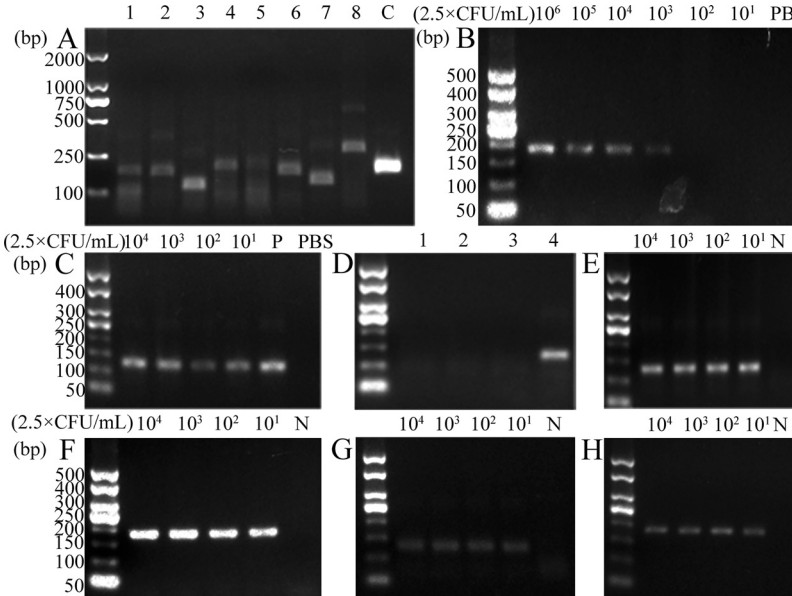

**FIG 5** Agarose gel electrophoresis images obtained through RPA and PCR methods. (A) Screening of eight primer pairs and positive quality control (1~8, C). Sensitivity of the (B) PCR and (C) RPA methods for serially diluted *S. aureus* cultures in PBS, 'P' represents positive sample of *S. aureus*. (D) Specificity of RPA to (1) *S. Typhimurium*, (2) *L. monocytogenes*, (3) *E. coli*, and (4) *S. aureus*. Sensitivity of the IMBs–RPA methods to detect *S. aureus* in spiked (E) milk and (F) pork samples, and sensitivity of IMBs–PCR in spiked (G) milk and (H) pork samples. 'N' represents a negative sample without *S. aureus*.

it requires 18 to 24 h for bacterial culture before detection, while IMB separation only takes 1 h to capture and enrich bacteria. Costa et al. (40) reported a 100% sensitive and specific real-time PCR method for the detection of methicillin resistance *S. aureus* compared with PCR, but samples needed to be inoculated onto 5% horse blood agar plates for 24 to 48 h before detection. Hu Y et al. (41) reported the detection limit of *S. aureus* was $1.4 \times 10^5$ CFU/mL by ELISA in spiked milk sample that was previously centrifuged and washed, while in our study the detection limit was $2.5 \times 10^1$ CFU/mL without centrifugation. As such, considerable research attention has been devoted to the development of efficient and rapid methods for the detection of *S. aureus* (42–44). Despite this, there has been little published research on the use of IMBs in conjunction with the RPA to capture and detect *S. aureus* strains in milk and pork products.

In this study, the prepared carboxyl $Fe_3O_4$ MBs were monodisperse and exhibited superparamagnetism, which are prerequisites for the isolation of pathogenic bacteria.

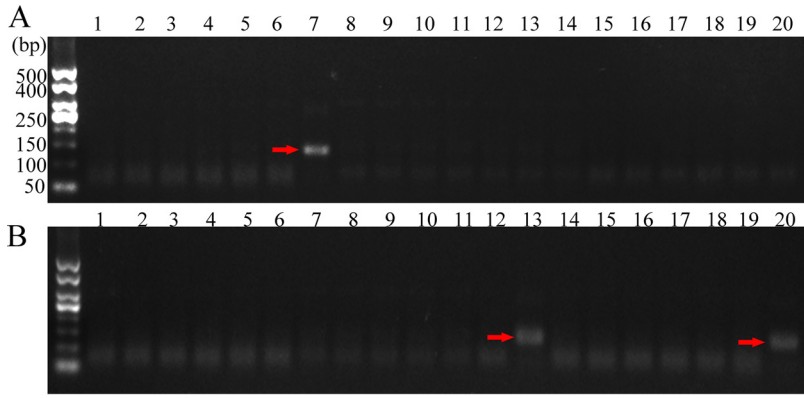

**FIG 6** Detection of *S. aureus* in (A) raw milk and (B) chilled pork samples through the IMBs–RPA method. The red arrows indicate the presence of *S. aureus* in the sample.

These IMBs showed high capture efficiency to *S. aureus* in spiked samples. Antibodies played a crucial role in the performance of IMBs. Compared with the monoclonal antibodies, the polyclonal antibodies, synthesized by a facile, low-cost method, had multiple epitope binding sites, which made the capture of the bacterial strains more conducive. Herein, the titer of the prepared high-purity polyclonal antibody was as high as 1:51 200. The capture performance of the IMBs was possibly affected by the particle size of the MBs. Skjerve et al. (45) reported that the capture efficiency of MBs with an average particle size of 2.8 $\mu$m was 35%, which was considerably lower than that of the $Fe_3O_4$ IMBs in this study. The larger specific surface area of nanosized MBs offers more available sites for the adsorption of the specific antibodies, thus increasing the chance of collision and binding between the MBs and the bacteria. Brandao et al. (46) also demonstrated that nanosized MBs have a higher capture efficiency than the micro-sized MBs. In this study, the monodisperse spherical MBs (average particle size = 200 nm) had more available sites to capture the antibodies, and thus exhibited a strong antibacterial performance. The mode of connection between the MBs and antibodies also affects the amount of the captured antibodies. Ding et al. (47) found that the number of dye molecules bound to cellulose nanocrystals through covalent bonds was five times higher than those bound through electrostatic adsorption. Therefore, the EDC-NHS chemical coupling method was employed herein to transform the carboxyl group on the surface of $Fe_3O_4$ MBs into intermediate active esters. The MBs and antibodies were conjugated through the covalent bonding of the NHS ester and amino groups on the surface of the antibodies.

The amplification efficiency of the RPA mainly depends on the selection of the target sequence and design of primers (38). The *nuc* gene was used to design RPA primers, which encodes a thermostable nuclease unique for the *S. aureus* strain (40). Eight primer pairs of *nuc* gene were designed and the *nuc*-2.1 oligonucleotides were determined to be the optimal pair. In addition, the temperature had no significant impact on the sensitivity. This agrees well with the results of previously reported studies, which reveal that the RPA reaction can be performed at human body temperature (40, 47). Herein, the reaction temperature was 39°C considering the TwisDx (UK) guidelines. More often, longer reaction times are necessary for low reaction temperatures. The RPA reaction carried out at 39°C using the optimal *nuc*-2.1 oligonucleotides had the best amplification efficiency and exhibited nonspecific amplification toward *S. Typhimurium*, *L. monocytogenes*, and *E. coli*. Therefore, the RPA was highly specific. The detection line of the RPA method for *S. aureus* was $2.5 \times 10^1$ CFU/mL, which was similar to that of multiobjective RPA combined lateral flow dipstick, with detection line of $2.6 \times 10^1$ CFU/mL (37). On the other hand, according to the national standard of the People's Republic of China (GB 29921-2021 limit of pathogenic bacteria in food), the acceptable level of *S. aureus* in meat products is 100 CFU/g (mL), and the maximum safe limit value is 1000 CFU/g (mL). Therefore, the sensitivity of the IMBs-RPA method established in our study can well meet the detection limit requirements in the national standard. After the isolation and enrichment of the *S. aureus* strains in artificially contaminated samples using IMBs, the detection line of PCR and RPA methods were both $2.5 \times 10^1$ CFU/mL. The IMBs specifically captured *S. aureus* in samples and separated them under the action of an external magnetic field, which realized the enrichment of *S. aureus*, and increased the template amounts that used for amplification, thus reduced the detection line and increased the sensitivity. So relative to the sensitivity of PCR method alone, a 100-fold increase was observed for IMBs-PCR and IMBs-RPA. Due to the high sensitivity of the RPA method, it is necessary to pay attention to the false-positive situation caused by aerosol contamination and to make a good partition between the reagent preparation, template addition, nucleic acid amplification, and product analysis areas during the RPA.

In this study, a novel and rapid detection method based on the combination of immunomagnetic separation and RPA for targeting the *nuc* gene of *S. aureus* was developed. The proposed method simplified the pretreatment procedures for pathogen detection in food samples. As such, short detection time, higher sensitivity, and

high specificity were achieved, which proves the potential of the novel route for food safety supervision.

## MATERIALS AND METHODS

**Reagents and materials.** Ferric chloride hexahydrate ($FeCl_3 \cdot 6H_2O$), sodium acetate (NaAc), ethylene glycol, polyethylene glycol (PEG-2000, $M_w = 2000$), and absolute ethanol were purchased from Sinopharm Chemical Reagent Co., Ltd. (Beijing, China). 1-Ethyl-3-(3-dimethylaminopropyl)-carbodiimide hydrochloride (EDC), N-hydroxysuccinimide (NHS), 2-Morpholinoethanesulfonic acid monohydrate (MES), bovine serum albumin (BSA) and Gram Stain kit were purchased from Solaibao Biological Technology Co., Ltd. (Beijing, China). Sodium azide ($NaN_3$) was purchased from Sigma-Aldrich (Shanghai, China). Seven point five percent Sodium Chloride Broth, Blood agar plate, Brain Heart Infusion Broth and Freeze-dried rabbit plasma were purchase from Qingdao Haibo Biology Company, China.

**Bacterial strains and culture media.** *S. aureus* (ATCC29740), *Salmonella* Typhimurium (*S. Typhimurium*, ATCC14028), *Escherichia coli* (*E. coli*, ATCC25922), and *Listeria monocytogenes* (*L. monocytogenes*, ATCC19115) were preserved in the laboratory. *L. monocytogenes* strains were cultured in tryptic soy broth (TSB, Qingdao Haibo Biology Company, China) at 37°C for 18 h. The other strains were cultured in Luria-Bertani broth (LB) at 37°C unless otherwise specified. The *S. aureus* strains captured by the IMBs was quantified at 37°C for 12 h using a Baird-Parker Agar (BPA, Qingdao Haibo Biology Company, China), and those of the other strains were determined using the LB agar.

**Raw milk and chilled pork samples.** The actual samples used in this study were collected in various districts of Beijing. Twenty raw milk samples were obtained from 10 different dairy farms, and chilled pork samples were obtained from 10 different farmers' markets in spring and summer.

**Synthesis of the carboxyl-$Fe_3O_4$ magnetic beads.** Monodisperse $Fe_3O_4$ magnetic beads were synthesized by solvothermal method. First, 3.51 g of $FeCl_3 \cdot 6H_2O$ and 7.46 g of NaAc were separately added to 40 mL of ethylene glycol under vigorous stirring for 30 min. Then, mixed the two together and added ethylene glycol to make final volume to 100 mL, the resulting precursor solution was stirred vigorously for another 3 h, and then transferred into a high-temperature reactor to react at 200°C for 5 h, followed by cooling to room temperature. Finally, the black target product was dried *in vacuo* at 60°C for 8 h after washing three times with ethanol and water alternately.

The morphology and structure of the samples were observed by field emission scanning electron microscopy (FE-SEM, Hitachi SU8010, Tokyo, Japan) and field-emission transmission electron microscopy (FE-TEM, JEOL, JEM-2100F, JEOL, Tokyo, Japan). The magnetic properties of the synthesized carboxyl-$Fe_3O_4$ MBs were measured at 25°C using a vibration sample magnetometer (VSM, Xinke, BKT-4500, Beijing, China). Lastly, to characterize the chemical structures of the samples, Fourier-transform infrared (FTIR) spectroscopy (Nicolet IS10, Madison, USA) was performed over the wave number range of 4000 to 400 $cm^{-1}$.

**Generation of the *S. aureus* polyclonal antibodies.** Two female New Zealand White rabbits (2.25 kg) were immunized with $5 \times 10^9$, $1 \times 10^{10}$, $1 \times 10^{10}$, $2 \times 10^{10}$, and $2 \times 10^{10}$ CFU (CFU) of heat-inactivated *S. aureus* on days 1, 4, 7, 14, and 21 of the experiment, respectively. Blood samples were collected from the heart 5 days after the last immunization. The anti-*S. aureus* polyclonal antisera were purified using a protein A antibody purification kit (Sangon Biotech [Shanghai] Co. Ltd., Shanghai, China). The concentration, purity, and titers of the purified antibodies were determined by bicinchoninic (BCA) protein assay (Solaibao Biological Technology Co., Ltd., Beijing, China), sodium dodecyl sulfate-polyacrylamide gel electrophoresis (SDS-PAGE) analysis, and ELISA.

**Preparation of the IMBs.** The carboxyl-$Fe_3O_4$ MBs were washed three times using MEST buffer (0.1 M MES, 0.05 vol% Tween 20, and pH 5.0) and then added 200 $\mu$L 0.1 M MES solution (pH 5.0) containing EDC (100 mg/mL) and NHS (100 mg/mL) successively. The MBs were activated through the rotary mixing of the resulting solution at 37°C for 1 h. Then, the components of the mixture were magnetically separated; the supernatant was discarded. The purified immunoglobulin (IgG) (0.5 mg) was added to the tube. The volume was made up to 500 $\mu$L through the addition of the MES. To obtain the IMBs, the resulting IgG-MB mixture was rotated for 1h at 37°C. The IMBs were blocked through incubation at 37°C for 1 h using 1 mL of 1% BSA (wt/vol) in 0.1 M MES buffer solution. Thereafter, they were washed thrice with 0.01 M phosphate-buffered saline (PBS) (pH 7.4). Finally, the IMBs (10 mg/mL) were resuspended in PBS containing 0.1% BSA (wt/vol) and 0.02% $NaN_3$ (wt/vol) and then stored at 4°C.

**Capture efficiency assay of the IMBs.** Freshly cultured *S. aureus* was diluted serially to prepare samples with cellular concentrations ranging from $2.5 \times 10^0$ to $2.5 \times 10^5$ CFU/mL in PBS. The IMBs were added to the serially diluted bacterial samples to a final concentration of 5 mg/mL and the blank control without the IMBs was set, followed by incubating the sample at 37 °C for 1 h with gentle rotation. The IMBs with the bound target bacteria were separated from the solution using a magnet, washed thrice with PBS to remove any unbound *S. aureus* strains, and then plated on the Baird–Parker agar medium to determine the capture efficiency of the IMBs on the target bacteria. Freshly cultured *S. Typhimurium*, *E. coli*, *L. monocytogenes*, and *S. aureus* were diluted to $10^4$ CFU/mL with PBS. To investigate the specificity of the IMBs toward *S. aureus*, the capture and plating processes were performed as described above. All the agar plates were incubated at 37°C for 12 to 16 h. Capture efficiency was calculated through the formula, $C = \frac{C_1}{C_0} \times 100\%$, where $C_0$ and $C_1$ represent the colony counts on the blank control and those bound to the IMBs, respectively.

**Primer design and performance detection of RPA.** The *nuc* gene was used to design RPA primers, which encodes a thermostable nuclease unique for *S. aureus* strain (40). The sequence of the *nuc* gene of *S. aureus* was downloaded from GenBank. Eight primer pairs (Table S1) for RPA were designed according to the recommendations of the TwisDx Assay Design Manual (TwisDx, Cambridge, UK), and

evaluated to estimate which one can provide the best amplification. RPA reactions were carried out with the Twist@basic kit (TwisDX, Cambridge, UK). RPA reaction system contains three enzymes, recombinases that bind single-stranded nucleic acids (oligonucleotide primers), single-stranded DNA-binding proteins (SSB), and strand-displaced DNA polymerases, these enzymes can affect the progress of DNA in polyacrylamide gel electrophoresis, lead to stripe dispersion. So the amplification products of RPA need to be purified by phenol-chloroform extraction and then visually detected via DNA gel electrophoresis. The RPA sensitivity was determined using DNA extraction of $2.5 \times 10^4$, $2.5 \times 10^3$, $2.5 \times 10^2$, $2.5 \times 10^1$, and $2.5 \times 10^0$CFU/mL *S. aureus* cells. Meanwhile, the DNA of *S. Typhimurium, E. coli, L. monocytogenes*, and *S. aureus* were used to study the specificity of the RPA method. The genomic DNA was extracted using a bacterial genomic extraction kit (AidlabBiotech Co., Ltd., Beijing) and detected using the RPA.

**Application of IMBs-RPA in spiked milk and pork.** Pasteurized milk and fresh pork samples purchased from supermarkets were found to be negative for *S. aureus* according to standard culture methods. *S. aureus* strain cultures with a final concentration of $2.5 \times 10^4$, $2.5 \times 10^3$, $2.5 \times 10^2$, $2.5 \times 10^1$, 10, 5, 2, and 1 CFU/mL were added to 10 mL of milk and pork homogenates. The *S. aureus* strains in such artificially contaminated samples were captured through immunomagnetic separation. Then, the genomic DNA of the *S. aureus* culture was extracted and detected using the RPA. To verify the results, PCR was performed with *Taq* DNA polymerase, and the reaction procedure was initial denaturation at 95°C for 5 min followed by 30 cycles denaturation at 95°C for 30 s, annealing at 51°C for 30 s, and extension at 72°C for 30 s. The *nuc*-1.2 oligonucleotides (Table S1) were used as specific primers for PCR.

**Detection of *S. aureus* in actual samples through the IMBs-RPA method.** The established IMBs-RPA method was used to detect 20 raw milk and chilled pork each from different dairy farms and farmers' market. The results of IMBs-RPA were also verified by PCR and *S. aureus* inspection procedure. Briefly, initial identification was carried out by the appearance of black single colonies on Baird-Parker plates and the transparent hemolytic rings on plates, and suspected colonies were further subjected to Gram staining and coagulase test (National Standard of the People's Republic of China, GB 4789.10-2016 *Staphylococcus aureus* Inspection Standard). Before the sample analysis, milk and pork samples were diluted or homogenized using PBS in a ratio of 1: 9. The amplification results were analyzed through DNA gel electrophoresis.

**Statistical analysis.** The experiments were performed in triplicates, and all measurements were conducted three times per experiment. Statistical analyses were performed using GraphPad software (LaJolla, CA, USA). Statistical significance was determined through the Student's *t* test and two-way analysis of variance (ANOVA). Differences were considered statistically significant if $P < 0.05$.

**Data availability.** All data generated or analyzed during this study are included in this published article (and its supplemental information files).

## SUPPLEMENTAL MATERIAL

Supplemental material is available online only.
**SUPPLEMENTAL FILE 1**, PDF file, 0.1 MB.

## ACKNOWLEDGMENTS

This work was funded by Beijing outstanding talent training program (2017000026833ZK18), and Beijing youth talent project (CIT &TCD201904052).

We have no conflicts of interest to declare.

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
