## [Reviewer comments · Microbiology Spectrum]

Microbiology Spectrum

Rapid detection of *Staphylococcus aureus* in milk and pork via immunomagnetic separation and recombinase polymerase amplification

Runan Cheng, Lei Li, Sihui Zhen, Honglei Liu, Zhouhui Wu, Yu Wang, and zhen wang

Corresponding Author(s): zhen wang, Beijing University of Agriculture

Review Timeline:

Submission Date:	June 15, 2022
Editorial Decision:	September 19, 2022
Revision Received:	October 4, 2022
Editorial Decision:	November 17, 2022
Revision Received:	November 18, 2022
Accepted:	February 4, 2023

Editor: Vincenzina Fusco

Reviewer(s): Disclosure of reviewer identity is with reference to reviewer comments included in decision letter(s). The following individuals involved in review of your submission have agreed to reveal their identity: Giuseppe Blaiotta (Reviewer #1); Laura Huber (Reviewer #3)

Transaction Report:

DOI: <https://doi.org/10.1128/spectrum.02249-22>

September 19, 2022

Dr. zhen wang
Beijing University of Agriculture
beijing
China

Re: Spectrum02249-22 (Rapid detection of *Staphylococcus aureus* in milk and pork via immunomagnetic separation and recombinase polymerase amplification)

Dear Dr. zhen wang:

Link Not Available

Sincerely,

Vincenzina Fusco

Journals Department
Reviewer comments:

Reviewer #1 (Comments for the Author):

Conclusions are well supported by obtained results

Minor remarks:

L 65-67: please insert some references

L100, L146: are the authors that 12-16 h of incubation are sufficient?

L118, L119, L127: check for spelling errors.

L221-227. the samples had to be analyzed also with the standard procedure (culturomics)

I suggest the authors to compare culturomics results with IMB-RPA approach

Reviewer #2 (Comments for the Author):

Review Cheng et al - Rapid detection of *S. aureus* in milk and pork via IMBs-RPA - sept 2022

This original work from Cheng et al aims at setting up a new approach to detect *S. aureus* in milk and pork samples. They used macrosized magnetic beads, coated with polyclonal antibodies against *S. aureus*, combined with a recombinase polymerase amplification, which is not more sensitive but more rapid compared to a classical PCR amplification.

The manuscript is written in correct English. I however can propose several points that should be fixed and/or answered by the authors to improved the paper.

Major comments:

- The discussion section is mainly restricted to a general repeat of the main results. For example, L233-237 please compare more precisely your results with sensitivity and rapidity of currently used methods.
- L220: please discuss why the IMBs-PCR displays 100 times higher specificity than PCR alone. Perhaps following concentration by IMB binding?
- Abstract I. 23 and Results I. 224-225: the authors have experienced their new detection assay on 20 milk samples and 20 chilled pork samples from "different dairy farms and farmers' market". It is interesting to see that some samples are indeed positive to *S. aureus*, which was confirmed with an IMBs-PCR approach. However, I think the authors could also test the same samples with another test currently used for the detection of *S. aureus* in such kind of samples in dairy industry and pork industry, in order to evaluate the gain in sensitivity on these "natural" samples with this new method, and even if this sensitivity was tested with specific inoculated samples. Moreover, as nothing is presented about the diversity of each set of 20 samples (eg type of agriculture (organic or not...), location (only China?, which regions?...), species of origin for the samples..., season of sample...), it is impossible to raise quantitative results such as claiming that 5% or 10% of the samples are positive. Please remove these statements.
- As unspecific amplifications could easily be observed using RPA, the authors must provide negative controls with milk and pork alone (before *S. aureus* addition), and must comment this in the manuscript. Perhaps this control is given in the lanes "N" of figure 4, but the authors have to explicit it in the legend. Legend of figure 4 is especially incomplete: please indicate the correspondence in panel A between each well and each pair of primers. Please specify which matrix and which oligo was used as "positive quality control". Please explain the correspondence panels E, F, G, H with milk and pork samples.
- L143-147: to study the specificity in the presence of various species, the authors could have assayed their approach in the presence simultaneously of *S. aureus*, *E. coli*, *S. Typhimurium*, and *L. monocytogenes*.
- L161-162: it could be of interest to know how does the approach works at different dilutions (1:2, 1:5, 1:20, 1:100...?). The authors should perform such kind of experiments.
- L167 and elsewhere: please could the authors explain why they qualify their samples from milk and pork as "clinical" samples? Are these samples isolated from sick animals? If not, please remove this qualifier.

Minor comments:

- L.31 as IMBs and RPA acronyms are not known from all microbiologists, I suggest to add they signification also in this section. *S. aureus* could also be a keyword?
- L.32 "Gram-positive" (Gram was a microbiologist and thus Gram always needs its first letter in upper case)
- L.52-53: these acronyms are not used later in the manuscript. As there are already many acronyms, I think these three ones could be removed (MDT, MH and MRI).
- L90: please add a space after "hydrochloride"
- L.95: "ATCC14028), *Escherichia coli*..." (please remove and)
- L.104: what does "EG" stands for?
- L106: please add a space after "200 {degree sign}C"
- L112: please add a space after "MBs" and after "25 {degree sign}C"
- L114: please add a space after "Fourier-transform"
- L118: please add a space after "*S. aureus*"
- L119: please add a space after "heart"
- L121: please add a space after "(Shanghai)"
- L127: please add a space before "successively"
- L131: "of MES" => "of MES buffer".
- L132: please correct: 1% BSA (w/v). idem L134
- L135: please add a space after "0.02"
- L148: please indicate here what does the nuc gene encodes, and why this gene has been chosen. This is only indicated in the discussion!...
- L151: "and evaluated to estimate which one can..."
- L153: please explain why a phenol chloroform extraction is needed here

- L164: please indicated if the extraction was performed with a bacterial genomic extraction kit?
- L164: "To verify the results, PCR amplifications were performed". And please indicate the different PCR temperature steps into brackets. It is not clear to assume that a PCR is performed at a specific temperature, whereas several steps are needed (not correct English).
- L165: please write nuc in italics as elsewhere in the manuscript.
- L165: "nuc 1.2" lacks a substantive, eg "nuc 1.2 oligonucleotides"
- L164-166: please specify which DNA polymerase was used in these PCR experiments.
- L183: VSM is not classical for a microbiologist, thus my advice is not to use an acronym and repeat the signification as it is only used at few occurrences in the text.
- L194: please add a space after "was"
- L212: Primer => primer
- L214: as expect from the figure legend, these sensitivities should be "1 x 10³ CFU/mL"... if 2.5, please correct this information in the figure legend.
- L216: please specify which concentration of bacteria was used, in number of CFU/mL, both in the text and in figure legend.
- L222: please remove "each"
- L223: one milk sample was tested (add was)
- L240: please remove figure citations, not needed in discussion section
- L263: please remove "according to the recommendations...". This is mat and methods information, and not expected in discussion!
- L264: "optimal primer" => "optimal pair »
- L266 : "body temperature" is not clear (depends on the organism!)
- L268: reaction times
- L272: which was similar to previously reported study.
- L272: please specify which approach was used in ref 32

Staff Comments:

Preparing Revision Guidelines

Please return the manuscript within 60 days; if you cannot complete the modification within this time period, please contact me. If you do not wish to modify the manuscript and prefer to submit it to another journal, please notify me of your decision immediately so that the manuscript may be formally withdrawn from consideration by Microbiology Spectrum.

Review Cheng et al – Rapid detection of *S aureus* in milk and pork via IMBs-RPA – sept 2022

This original work from Cheng et al aims at setting up a new approach to detect *S. aureus* in milk and pork samples. They used macrosized magnetic beads, coated with polyclonal antibodies against *S. aureus*, combined with a recombinase polymerase amplification, which is not more sensitive but more rapid compared to a classical PCR amplification.

The manuscript is written in correct English. I however can propose several points that should be fixed and/or answered by the authors to improved the paper.

Major comments:

- The discussion section is mainly restricted to a general repeat of the main results. For example, L233-237 please compare more precisely your results with sensitivity and rapidity of currently used methods.
- L220: please discuss why the IMBs-PCR displays 100 times higher specificity than PCR alone. Perhaps following concentration by IMB binding?
- Abstract l. 23 and Results l. 224-225: the authors have experienced their new detection assay on 20 milk samples and 20 chilled pork samples from “different dairy farms and farmers’ market”. It is interesting to see that some samples are indeed positive to *S. aureus*, which was confirmed with an IMBs-PCR approach. However, I think the authors could also test the same samples with another test currently used for the detection of *S. aureus* in such kind of samples in dairy industry and pork industry, in order to evaluate the gain in sensitivity on these “natural” samples with this new method, and even if this sensitivity was tested with specific inoculated samples. Moreover, as nothing is presented about the diversity of each set of 20 samples (eg type of agriculture (organic or not...), location (only China?, which regions?...), species of origin for the samples..., season of sample...), it is impossible to raise quantitative results such as claiming that 5% or 10% of the samples are positive. Please remove these statements.
- As unspecific amplifications could easily be observed using RPA, the authors must provide negative controls with milk and pork alone (before *S. aureus* addition), and must comment this in the manuscript. Perhaps this control is given in the lanes “N” of figure 4, but the authors have to explicit it in the legend. Legend of figure 4 is especially incomplete: please indicate the correspondence in panel A between each well and each pair of primers. Please specify which matrix and which oligo was used as “positive quality control”. Please explain the correspondence panels E, F, G, H with milk and pork samples.
- L143-147: to study the specificity in the presence of various species, the authors could have assayed their approach in the presence **simultaneously** of *S. aureus*, *E. coli*, *S. Typhimurium*, and *L. monocytogenes*.
- L161-162: it could be of interest to know how does the approach works at different dilutions (1:2, 1:5, 1:20, 1:100...?). The authors should perform such kind of experiments.

- L167 and elsewhere: please could the authors explain why they qualify their samples from milk and pork as “clinical” samples? Are these samples isolated from sick animals? If not, please remove this qualifier.

Minor comments:

- L.31 as IMBs and RPA acronyms are not known from all microbiologists, I suggest to add they signification also in this section. *S. aureus* could also be a keyword?
- L.32 “Gram-positive” (Gram was a microbiologist and thus Gram always needs its first letter in upper case)
- L.52-53: these acronyms are not used later in the manuscript. As there are already many acronyms, I think these three ones could be removed (MDT, MH and MRI).
- L90: please add a space after “hydrochloride”
- L.95: “ATCC14028), *Escherichia coli*...” (please remove and)
- L.104: what does “EG” stands for?
- L106: please add a space after “200 °C”
- L112: please add a space after “MBs” and after “25 °C”
- L114: please add a space after “Fourier-transform”
- L118: please add a space after “*S. aureus*”
- L119: please add a space after “heart”
- L121: please add a space after “(Shanghai)”
- L127: please add a space before “successively”
- L131: “of MES” => “of MES buffer”.
- L132: please correct: 1% BSA (w/v). idem L134
- L135: please add a space after “0.02”
- L148: please indicate here what does the *nuc* gene encodes, and why this gene has been chosen. This is only indicated in the discussion!...
- L151: “and evaluated to estimate which one can...”
- L153: please explain why a phenol chloroform extraction is needed here
- L164: please indicated if the extraction was performed with a bacterial genomic extraction kit?
- L164: “To verify the results, PCR amplifications were performed”. And please indicate the different PCR temperature steps into brackets. It is not clear to assume that a PCR is performed at a specific temperature, whereas several steps are needed (not correct English).
- L165: please write *nuc* in italics as elsewhere in the manuscript.
- L165: “nuc 1.2” lacks a substantive, eg “nuc 1.2 oligonucleotides”
- L164-166: please specify which DNA polymerase was used in these PCR experiments.
- L183: VSM is not classical for a microbiologist, thus my advice is not to use an acronym and repeat the signification as it is only used at few occurrences in the text.
- L194: please add a space after “was”
- L212: Primer => primer
- L214: as expect from the figure legend, these sensitivities should be “1 x 10³ CFU/mL” ... if 2.5, please correct this information in the figure legend.
- L216: please specify which concentration of bacteria was used, in number of CFU/mL, both in the text and in figure legend.
- L222: please remove “each”
- L223: one milk sample was tested (add was)

- L240: please remove figure citations, not needed in discussion section
- L263: please remove "according to the recommendations...". This is mat and methods information, and not expected in discussion!
- L264: "optimal primer" => "optimal pair »
- L266 : "body temperature" is not clear (depends on the organism!)
- L268: reaction times
- L272: which was similar to previously reported study.
- L272: please specify which approach was used in ref 32

Reviewer #1 (Comments for the Author):

Conclusions are well supported by obtained results

Minor remarks:

L 65-67: please insert some references

Author : Thank you for your reminding, references have been added in the revised manuscript.

L100, L146: are the authors that 12-16 h of incubation are sufficient?

Author : For *S. aureus* strain that kept in the laboratory, single colonies of 1-2 mm in diameter were observed in Baird-Parker Agar for 12-16 h. To isolate *S. aureus* from clinical samples, 18 h to 24 h culture is required.

L118, L119, L127: check for spelling errors.

Author : Thanks for your reminding. We have revised the writing specifications of all manuscripts.

L221-227. the samples had to be analyzed also with the standard procedure (culturomics) I suggest the authors to compare culturomics results with IMB-RPA approach

Author: Thank you for your suggestion. For the detection of actual samples, we simultaneously used the IMBS-RPA method that established in this study and the inspection procedure of the National Standard of the People's Republic of China (GB 4789.10-2016 *Staphylococcus aureus* Inspection Standard). According to standard procedure, one case of raw milk sample and two cases of pork samples among 20 actual samples showed black *S. aureus* single colonies on Baird-Parker plate and hemolysis on blood plate, and the biochemical identification was positive for haptozyme, which was consistent with the detection results of IMBs-RPA method that established in this study.

Reviewer #2 (Comments for the Author):

Review Cheng et al - Rapid detection of *S aureus* in milk and pork via IMBs-RPA - sept 2022

This original work from Cheng et al aims at setting up a new approach to detect *S. aureus* in milk and pork samples. They used macrosized magnetic beads, coated with polyclonal antibodies against *S. aureus*, combined with a recombinase polymerase amplification, which is not more sensitive but more rapid compared to a classical PCR amplification.

The manuscript is written in correct English. I however can propose several points that should be fixed and/or answered by the authors to improved the paper.

Major comments:

- The discussion section is mainly restricted to a general repeat of the main results. For example, L233-237 please compare more precisely your results with sensitivity and rapidity of currently used methods.

Author: Thank you for your suggestion. The discussion section has been revised.

- L220: please discuss why the IMBs-PCR displays 100 times higher specificity than PCR

alone. Perhaps following concentration by IMB binding?

Author : The immunomagnetic beads specifically captured *Staphylococcus aureus* in samples, and separated them under the action of external magnetic field, which realized the enrichment of *S. aureus*, and increased the template amount that used for PCR, thus reduce the detection line and increase the sensitivity. while without IMBs, the detection line was 100 times higher when using PCR alone.

Abstract I. 23 and Results I. 224-225: the authors have experienced their new detection assay on 20 milk samples and 20 chilled pork samples from "different dairy farms and farmers' market". It is interesting to see that some samples are indeed positive to *S. aureus*, which was confirmed with an IMBs-PCR approach. However, I think the authors could also test the same samples with another test currently used for the detection of *S. aureus* in such kind of samples in dairy industry and pork industry, in order to evaluate the gain in sensitivity on these "natural" samples with this new method, and even if this sensitivity was tested with specific inoculated samples. Moreover, as nothing is presented about the diversity of each set of 20 samples (eg type of agriculture (organic or not...), location (only China?, which regions?...), species of origin for the samples..., season of sample...), it is impossible to raise quantitative results such as claiming that 5% or 10% of the samples are positive. Please remove these statements.

Author : Thank you for your suggestion. Raw milk and chilled pork samples used in this study were collected in various districts of Beijing. Twenty raw milk samples were obtained from 10 different dairy farms, and chilled pork samples were obtained from 10 different farmers' markets in spring and summer.

For the detection of actual samples, we simultaneously used the IMBS-RPA method that established in this study and the inspection procedure of the National Standard of the People's Republic of China (GB 4789.10-2016 *Staphylococcus aureus* Inspection Standard). According to standard procedure, one case of raw milk sample and two cases of pork samples among 20 actual samples showed black *S. aureus* single colonies on Baird-Parker plate and hemolysis on blood plate, and the biochemical identification was positive for haptozyme, which was consistent with the detection results of IMBS-RPA method that established in this study.

In the revised manuscript, we have removed the positive rate and only show the number of positive samples.

- As unspecific amplifications could easily be observed using RPA, the authors must provide negative controls with milk and pork alone (before *S. aureus* addition), and must comment this in the manuscript. Perhaps this control is given in the lanes "N" of figure 4, but the authors have to explicit it in the legend. Legend of figure 4 is especially incomplete: please indicate the correspondence in panel A between each well and each pair of primers. Please specify which matrix and which oligo was used as "positive quality control". Please explain the correspondence panels E, F, G, H with milk and pork samples.

Author : Thank you for your suggestion. The 'N' in Figure 4 is indeed a negative control

without *Staphylococcus aureus*. According to your suggestion, we re-annotated the figure legends in detail.

- L143-147: to study the specificity in the presence of various species, the authors could have assayed their approach in the presence simultaneously of *S. aureus*, *E. coli*, *S. Typhimurium*, and *L. monocytogenes*.

Authors: Thank you for your constructive suggestions. We made a new specificity test of *Staphylococcus aureus* IMBs for the mixed samples, and the bacteria amount of *S. aureus*, *E. coli*, *S. Typhimurium*, and *L. Monocytogenes* was 1:1:1: 1, each type of bacteria accounts for 25%. According to the plate count results, the capture rate of IMBs was only 21%, which indicated that the IMBs can only effectively capture *S. aureus* in the presence of other bacteria.

- L161-162: it could be of interest to know how does the approach works at different dilutions (1:2, 1:5, 1:20, 1:100...?). The authors should perform such kind of experiments.

Authors: This is indeed a good question and idea. According to your suggestion, we re-detected the detection line of *S. aureus* in artificial simulated samples, and the final concentration of bacteria was set as 50 CFU/mL, 25 CFU/mL, 10 CFU/mL, 5 CFU/mL, 2 CFU/mL, 1 CFU/mL. The results showed that no positive results could be detected below 10 CFU/mL. Which indicated that the lowest detection line of IMBS-RPA method that established in this study is 10 CFU/mL, which is the same order of magnitude as the results presented in the manuscript (2.5×10^1 CFU/mL). According to the national standard of the People's Republic of China (GB 29921-2021 limit of pathogenic bacteria in food), the acceptable level of *S. aureus* in meat products is 100 CFU/g (mL), and the maximum safe limit value is 1000 CFU/g (mL). Therefore, The sensitivity (10 CFU/ mL) of the IMBS-RPA method established in our study can well meet the detection limit requirements in the national standard.

- L167 and elsewhere: please could the authors explain why they qualify their samples from milk and pork as "clinical" samples? Are these samples isolated from sick animals? If not, please remove this qualifier.

Authors: Thank you for your reminding. The samples were not collected from sick animals, but from the raw cow milk of dairy farms and chilled porks were purchased from farmers' markets. In the revised manuscript, "clinical" samples has been changed to "actual samples".

Minor comments:

- L.31 as IMBs and RPA acronyms are not known from all microbiologists, I suggest to add they signification also in this section. *S. aureus* could also be a keyword?

Authors: Thanks for your reminding. *Staphylococcus aureus* has been added to the keywords in the revised manuscript.

- L.32 "Gram-positive" (Gram was a microbiologist and thus Gram always needs its first letter in upper case

Authors: Thank you for your reminding. gram-positive has been changed to "Gram-positive"

in the revised manuscript.

- L.52-53: these acronyms are not used later in the manuscript. As there are already many acronyms, I think these three ones could be removed (MDT, MH and MRI).

Authors: Thanks for your reminding. These acronyms have been removed from the revised manuscript.

- L90: please add a space after "hydrochloride"

Authors: Thank you for your reminding, spaces have been added in the revised manuscript.

- L.95: "ATCC14028), Escherichia coli..." (please remove and)

Authors: Thanks for your reminding. "and" has been removed from the revised manuscript.

- L.104: what does "EG" stands for?

Author : Thank you for your reminding, "EG" stands forethylene glycol

- L106: please add a space after "200 {degree sign}C"

Authors: Thank you for your reminding, spaces have been added in the revised manuscript.

- L112: please add a space after "MBs" and after "25 {degree sign}C"

Authors: Thank you for your reminding, spaces have been added in the revised manuscript.

- L114: please add a space after "Fourier-transform"

Authors: Thank you for your reminding, spaces have been added in the revised manuscript.

- L118: please add a space after "S. aureus"

Authors: Thank you for your reminding, spaces have been added in the revised manuscript.-

L119: please add a space after "heart"

Authors: Thank you for your reminding, spaces have been added in the revised manuscript.

- L121: please add a space after "(Shanghai)"

Authors: Thank you for your reminding, spaces have been added in the revised manuscript.

- L127: please add a space before "successively"

Authors: Thank you for your reminding, spaces have been added in the revised manuscript.

- L131: "of MES" => "of MES buffer".

Authors: Thank you for your reminding, it has been revised.

- L132: please correct: 1% BSA (w/v). idem L134

Authors: Thank you for your reminding, it has been corrected.

- L135: please add a space after "0.02"

Authors: Thank you for your reminding, spaces have been added in the revised manuscript.

- L148: please indicate here what does the nuc gene encodes, and why this gene has been chosen. This is only indicated in the discussion!...

Author : Thanks for your suggestion. We have added the reason for selecting *nuc* gene and the encode product. The *nuc* gene was used to design RPA primers, which encodes for a thermostable nuclease unique for *S. aureus* strain.

- L151: "and evaluated to estimate which one can..."

Authors: Thank you for your reminding, the description has been revised.

- L153: please explain why a phenol chloroform extraction is needed here

Author : RPA reaction system contains three enzymes, recombinases that bind single-stranded nucleic acids (oligonucleotide primers), single-stranded DNA-binding proteins (SSB), and strand-displaced DNA polymerases, these enzymes can affect the progress of DNA in polyacrylamide gel electrophoresis, lead to stripe dispersion. So we need to use phenol chloroform extraction to separate DNA.

- L164: please indicated if the extraction was performed with a bacterial genomic extraction kit?

Author : The bacterial genome extraction kit was used in the study. The name and manufacturer of the kit have been added in the revised manuscript.

- L164: "To verify the results, PCR amplifications were performed". And please indicate the different PCR temperature steps into brackets. It is not clear to assume that a PCR is performed at a specific temperature, whereas several steps are needed (not correct English).

Authors: The enzyme used in the PCR reaction was Taq DNA polymerase, and the reaction procedure was predenaturation at 95 °C for 5 min, denaturation at 95 °C for 30 s, renaturation at 51 °C for 30 s, and extension at 72 °C for 30 s.

- L165: please write nuc in italics as elsewhere in the manuscript.

Authors: Thanks for your reminding, it has been revised.

- L165: "nuc 1.2" lacks a substantive, eg "nuc 1.2 oligonucleotides"

Authors: Thanks for your reminding, it has been revised.

- L164-166: please specify which DNA polymerase was used in these PCR experiments.

Authors: Thanks for your reminding, Taq DNA polymerase was added.

- L183: VSM is not classical for a microbiologist, thus my advice is not to use an acronym and repeat the signification as it is only used at few occurrences in the text.

Authors: Thanks for your reminding, it has been revised.

- L194: please add a space after "was"

Authors: Thanks for your reminding, space has been added.

- L212: Primer => primer

Authors: Thanks for your reminding, it has been revised.

- L214: as expect from the figure legend, these sensitivities should be "1 x 10³ CFU/mL"... if 2.5, please correct this information in the figure legend.

Authors: Thanks for your reminding, it has been revised.

- L216: please specify which concentration of bacteria was used, in number of CFU/mL, both in the text and in figure legend.

Authors: Thanks for your reminding, it has been revised in the text and figure.

- L222: please remove "each"

Authors: Thanks for your reminding, "each" was removed.

- L223: one milk sample was tested (add was)

Authors: Thanks for your reminding, " was " has been added.

- L240: please remove figure citations, not needed in discussion section

Authors: Thanks for your reminding, these have been removed.

- L263: please remove "according to the recommendations...". This is mat and methods information, and not expected in discussion!

Authors: Thanks for your reminding, it has been revised

- L264: "optimal primer" => "optimal pair »

Authors: Thanks for your reminding, it has been revised.

- L266 : "body temperature" is not clear (depends on the organism!)

Author : Thanks for your reminding, it has been revised to human body temperature.

- L268: reaction times

Authors: Thanks for your reminding, it has been revised

- L272: which was similar to previously reported study.

Authors: Thanks for your reminding, it has been revised

- L272: please specify which approach was used in ref 32

Authors: Thanks for your reminding, it has been added in the revised manuscript. The multi-objective recombinase polymerase amplification (RPA) combined with a lateral flow dipstick (LFD) was used in this reference.

Reviewer 3

This paper shows a novel method for rapid identification of *S. aureus* in milk and meat samples. It has a high impact and importance on the field. However, major review needs to be done both in the written language and methods/conclusions.

Specifically, sensitivity and specificity analysis need to be performed given the importance of these terms in the paper.

Abstract

Line 23: The percentages are not correct.

Authors: Thanks for your reminding. In the revised manuscript, we have removed the positive rate and only show the number of positive samples.

Line 26: I don't understand what bacterial separation means in this context.

Authors: Bacteria separation refers to using culture medium to separate *Staphylococcus aureus* from a sample.

Line 28: complex matrix of what? Be more specific. Complex microbiome?

Authors: Complex matrix refers to milk and meat samples, the presence of fat, proteins and other organic substances in these samples may affect the test results.

Introduction

Line 33: delete dusts, and write cavities and skins in singular. Revise the plural vs singular terms throughout the text. In a lot of times, the plural is not necessary.

Authors: Thanks for your reminding. We have paid attention to the singular and plural problems in the revised manuscript.

Line 53: spaces before the ()

Authors: Thank you for your reminding, a space was added before the parentheses.

Line 62. Punctuation after the reference 26

Authors: Thank you for your reminding, the punctuation mark after reference 26 have been removed.

Line 68. Specify what sample matrix mean.

Authors: The actual test samples are meat products, dairy products, etc., the presence of fat, proteins and other organic matter in these samples may affect the test results.

Line 72. Reference missing

Authors: Thank you for your reminding, we have added reference in the revised manuscript.

Line 85. qPCR?

Authors: It's a regular PCR, not a qPCR.

Line 106. Missing space.

Authors: Thank you for your reminding, the space has been added in the revised manuscript.

Line 112. Missing space.

Authors: Thank you for your reminding, the space has been added in the revised manuscript.

Line 118. Missing space.

Authors: Thank you for your reminding, the space has been added in the revised manuscript.

Line 119. Missing space. A lot of missing spaces in different part of the paper. Revise thoroughly.

Authors: Thanks for your reminding, we have revised the full manuscript.

Line 134: I am not sure what the wt% means.

Authors: I am very sorry for our writing error, it should be 1% BSA (w/v) instead of wt%.

Result

Line 202-210. I am not sure you can state there is a high sensitivity and specificity here without statistically calculating those rates. I would suggest using "high capture efficiency" then.

Author : Thank you for your reminding, the high sensitivity means that the immune magnetic beads can capture the target at a low bacteria concentration, which have a high capture rate. According to your advice, high sensitivity has been revised to a high capture feature.

Line 212. Change the "Primer" to "primer"

Authors: Thanks for your reminding. "Primer" has been modified to "primer".

Line 216-217. Again here, I am not sure you can state there is a high sensitivity and specificity.

Authors: In the artificial simulated meat and milk samples, the lowest detection line of RPA method for *Staphylococcus aureus* was 2.5×10 CFU/mL, and the detection line of PCR was 2.5×10^3 CFU/mL. Therefore, we said that RPA method had higher sensitivity. The established RPA method of *S. aureus* failed to amplify a positive signal for *Escherichia coli*, *Salmonella* and *Listeria*, so we said that the RPA method was highly specific.

On the results section, specify which groups were compared and the statistical significance of the results. I see the comparisons are made in the figures but not mentioned in the text.

Author : Thank you for your suggestion. The statistical difference has been specifically expressed in the revised manuscript.

Discussion

Line 229-230. Be more specific, which adverse effects?

Authors: Thank you for your reminding. The more specific effects had been added in the revised manuscript.

Line 240-241. Again here, either explain better how you got to say high sensitivity and specificity or rephrase. I did not see any analysis on this aspect.

Authors: Thank you for your reminding, we have changed the sensitivity and specificity here to capture features.

Line 244-245: what is this number? 1:51,200?

Authors: 1:51,200 is the antibody titer.

Last paragraph needs to be updated depending on sensitivity and specificity analysis are performed or not. My suggestion would be to compare sensitivity and specificity between traditional methods and this novel method.

Authors: For the detection of actual samples, we simultaneously used the IMBS-RPA method that established in this study and the inspection procedure of the National Standard of the People's Republic of China (GB 4789.10-2016 *Staphylococcus aureus* Inspection Standard). According to standard procedure, one case of raw milk sample and two cases of pork samples among 20 actual samples showed black *S. aureus* single colonies on Baird-Parker plate and hemolysis on blood plate, and the biochemical identification was positive for haptozyme, which was consistent with the detection results of IMBS-RPA method that established in this study.

Figure legends:

figures should be able to stand alone. Describe figures better in the legends, including the acronyms

Authors: Thank you for your suggestion. The figure legends have been modified in the revised manuscript.

November 17, 2022

Dr. zhen wang
Beijing University of Agriculture
beijing
China

Re: Spectrum02249-22R1 (Rapid detection of *Staphylococcus aureus* in milk and pork via immunomagnetic separation and recombinase polymerase amplification)

Dear Dr. zhen wang:

I had a hard time in finding available reviewers. However, I also read your revised manuscript and do agree with Reviewer's suggestions to significantly modify your manuscript.

Link Not Available

Sincerely,

Vincenzina Fusco

Journals Department
Reviewer comments:

Reviewer #2 (Comments for the Author):

See attached file

Staff Comments:

Preparing Revision Guidelines

Please return the manuscript within 60 days; if you cannot complete the modification within this time period, please contact me. If you do not wish to modify the manuscript and prefer to submit it to another journal, please notify me of your decision immediately so that the manuscript may be formally withdrawn from consideration by Microbiology Spectrum.

The work from Cheng et al has been significantly improved following reviewers' recommendations. aims at setting up a new approach to detect *S. aureus* in milk and pork samples.

However, in my opinion there are still some points to be fixed before publication. Especially, it is a little bit annoying to see that several recommendations were not correctly whereas the authors claim they were. Please double check each point. Moreover, in most cases when a reviewer requests an explanation, it is generally assumed that the corresponding info has to be added in the manuscript.

- L220: please discuss why the IMBs-PCR displays 100 times higher specificity than PCR alone. Perhaps following concentration by IMB binding?

Author : The immunomagnetic beads specifically captured *Staphylococcus aureus* in samples, and separated them under the action of external magnetic field, which realized the enrichment of *S. aureus*, and increased the template amount that used for PCR, thus reduce the detection line and increase the sensitivity. while without IMBs, the detection line was 100 times higher when using PCR alone.

⇒ **Please add this information in the discussion section**

Abstract I. 23 and Results I. 224-225: the authors have experienced their new detection assay on 20 milk samples and 20 chilled pork samples from "different dairy farms and farmers' market". It is interesting to see that some samples are indeed positive to *S. aureus*, which was confirmed with an IMBs-PCR approach. However, I think the authors could also test the same samples with another test currently used for the detection of *S. aureus* in such kind of samples in dairy industry and pork industry, in order to evaluate the gain in sensitivity on these "natural" samples with this new method, and even if this sensitivity was tested with specific inoculated samples. Moreover, as nothing is presented about the diversity of each set of 20 samples (eg type of agriculture (organic or not...), location (only China?, which regions?...), species of origin for the samples..., season of sample...), it is impossible to raise quantitative results such as claiming that 5% or 10% of the samples are positive. Please remove these statements.

Author : Thank you for your suggestion. Raw milk and chilled pork samples used in this study were collected in various districts of Beijing. Twenty raw milk samples were obtained from 10 different dairy farms, and chilled pork samples were obtained from 10 different farmers' markets in spring and summer.

⇒ **Please add this info in the Mat and Methods section**

For the detection of actual samples, we simultaneously used the IMBS-RPA method that established in this study and the inspection procedure of the National Standard of the People's Republic of China (GB 4789.10-2016 *Staphylococcus aureus* Inspection Standard). According to standard procedure, one case of raw milk sample and two cases of pork samples among 20 actual samples showed black *S. aureus* single colonies on Baird-Parker plate and hemolysis on blood plate, and the biochemical identification was positive for

haptozyme, which was consistent with the detection results of IMBS-RPA method that established in this study.

In the revised manuscript, we have removed the positive rate and only show the number of positive samples.

⇒ **Thank you**

- L143-147: to study the specificity in the presence of various species, the authors could have assayed their approach in the presence simultaneously of *S. aureus*, *E. coli*, *S. Typhimurium*, and *L. monocytogenes*.

Authors: Thank you for your constructive suggestions. We made a new specificity test of *Staphylococcus aureus* IMBs for the mixed samples, and the bacteria amount of *S. aureus*, *E. coli*, *S. Typhimurium*, and *L. Monocytogenes* was 1:1:1: 1, each type of bacteria accounts for 25%. According to the plate count results, the capture rate of IMBs was only 21%, which indicated that the IMBs can only effectively capture *S. aureus* in the presence of other bacteria.

⇒ **In the presence of other bacteria, the capture efficiency of *S. aureus* dramatically decreased from about 90% to about 20%. Please discuss this decrease in the discussion section.**

- L161-162: it could be of interest to know how does the approach works at different dilutions (1:2, 1:5, 1:20, 1:100...?). The authors should perform such kind of experiments.

Authors: This is indeed a good question and idea. According to your suggestion, we re-detected the detection line of *S. aureus* in artificial simulated samples, and the final concentration of bacteria was set as 50 CFU/mL, 25 CFU/mL, 10 CFU/mL, 5 CFU/mL, 2 CFU/mL, 1 CFU/mL. The results showed that no positive results could be detected below 10 CFU/mL. Which indicated that the lowest detection line of IMBS-RPA method that established in this study is 10 CFU/mL, which is the same order of magnitude as the results presented in the manuscript (2.5×10^1 CFU/mL).

⇒ **Please add this in the corresponding Mat and Methods section. I do not remember that it was added.**

- L153: please explain why a phenol chloroform extraction is needed here

Author : RPA reaction system contains three enzymes, recombinases that bind single-stranded nucleic acids (oligonucleotide primers), single-stranded DNA-binding proteins (SSB), and strand-displaced DNA polymerases, these enzymes can affect the progress of DNA in polyacrylamide gel electrophoresis, lead to stripe dispersion. So we need to use phenol chloroform extraction to separate DNA.

⇒ **Please add this info in the Mat and methods section**

- L263: please remove "according to the recommendations...". This is mat and methods

information, and not expected in discussion!

⇒ **This modification has not been done. Please correct!**

- L268: reaction times

Authors: Thanks for your reminding, it has been revised

⇒ **This modification has not been done. Please correct!**

Line 68 reviewer 3 asked about the meaning of sample matrix.

⇒ **Please add this information in the manuscript.**

Other minor points:

- First page: "Author order was determined on the basis of contribution".
- ⇒ **This is obvious and should always be the case. This sentence is not required here.**
- L54: was based **on** microbial culturing
- L55: **id**entification
- L57: were commonly used => are commonly used
- L58: **are** also used.
- L163: **were** used as specific primers for PCR.
- L168: procedure. **Briefly** (please make a new sentence)
- L170: to Gram straining **and** coagulase test
- L206: CFU/**mL**. **When** there were only... (please make a new sentence)
- L209: CFU/**mL**. **The** capture efficiency... (please make a new sentence)
- L211: Interestingly,
- L211-212: for **a** mixed sample **with** the bacteria...
- L212-213: this is not correct English, please rephrase
- L236: remove "that"
- L237: shorten => shortened
- L248: by ELISA **in** spiked milk sample **that was previously** centrifuged...
- L249: CFU/**mL** **without centrifugation**.
- L251: compared **with** PCR, but samples needed to **be** inoculated onto...

The work from Cheng et al has been significantly improved following reviewers' recommendations. aims at setting up a new approach to detect *S. aureus* in milk and pork samples.

However, in my opinion there are still some points to be fixed before publication. Especially, it is a little bit annoying to see that several recommendations were not correctly whereas the authors claim they were. Please double check each point. Moreover, in most cases when a reviewer requests an explanation, it is generally assumed that the corresponding info has to be added in the manuscript.

- L220: please discuss why the IMBs-PCR displays 100 times higher specificity than PCR alone. Perhaps following concentration by IMB binding?

Author : The immunomagnetic beads specifically captured *Staphylococcus aureus* in samples, and separated them under the action of external magnetic field, which realized the enrichment of *S. aureus*, and increased the template amount that used for PCR, thus reduce the detection line and increase the sensitivity. while without IMBs, the detection line was 100 times higher when using PCR alone.

⇒ **Please add this information in the discussion section**

Authors: Thanks for your reminding, this part has been added in the revised manuscript.

Abstract I. 23 and Results I. 224-225: the authors have experienced their new detection assay on 20 milk samples and 20 chilled pork samples from "different dairy farms and farmers' market". It is interesting to see that some samples are indeed positive to *S. aureus*, which was confirmed with an IMBs-PCR approach. However, I think the authors could also test the same samples with another test currently used for the detection of *S. aureus* in such kind of samples in dairy industry and pork industry, in order to evaluate the gain in sensitivity on these "natural" samples with this new method, and even if this sensitivity was tested with specific inoculated samples. Moreover, as nothing is presented about the diversity of each set of 20 samples (eg type of agriculture (organic or not...), location (only China?, which regions?...), species of origin for the samples..., season of sample...), it is impossible to raise quantitative results such as claiming that 5% or 10% of the samples are positive. Please remove these statements.

Author : Thank you for your suggestion. Raw milk and chilled pork samples used in this study were collected in various districts of Beijing. Twenty raw milk samples were obtained from 10 different dairy farms, and chilled pork samples were obtained from 10 different farmers' markets in spring and summer.

⇒ **Please add this info in the Mat and Methods section**

Authors: Thanks for your reminding, this part has been added in the revised manuscript.

For the detection of actual samples, we simultaneously used the IMBS-RPA method that established in this study and the inspection procedure of the National Standard of the People's Republic of China (GB 4789.10-2016 *Staphylococcus aureus* Inspection Standard). According to standard procedure, one case of raw milk sample and two cases of pork samples among 20 actual samples showed black *S. aureus* single colonies on Baird-Parker plate and hemolysis on blood plate, and the biochemical identification was positive for

haptozyme, which was consistent with the detection results of IMBS-RPA method that established in this study.

In the revised manuscript, we have removed the positive rate and only show the number of positive samples.

⇒ **Thank you**

- L143-147: to study the specificity in the presence of various species, the authors could have assayed their approach in the presence simultaneously of *S. aureus*, *E. coli*, *S. Typhimurium*, and *L. monocytogenes*.

Authors: Thank you for your constructive suggestions. We made a new specificity test of *Staphylococcus aureus* IMBs for the mixed samples, and the bacteria amount of *S. aureus*, *E. coli*, *S. Typhimurium*, and *L. Monocytogenes* was 1:1:1: 1, each type of bacteria accounts for 25%. According to the plate count results, the capture rate of IMBs was only 21%, which indicated that the IMBs can only effectively capture *S. aureus* in the presence of other bacteria.

⇒ **In the presence of other bacteria, the capture efficiency of *S. aureus* dramatically decreased from about 90% to about 20%. Please discuss this decrease in the discussion section.**

Authors: I'm very sorry that we didn't explain clearly. In fact, among the mixed bacteria sample, the capture rate of IMBs for *S. aureus* did not decrease, and the capture rate was 84%. Due to the bacteria amount proportions of *S. aureus*, *E. coli*, *S. typhimurium*, and *L. monocytogenes* in the mixed bacteria sample were all 25%, so the captured *S. aureus* (84%) accounted for 21% of the total number of bacteria in the mixed sample ($84\% \times 0.25=21\%$). In the revised manuscript, we have explained the result.

- L161-162: it could be of interest to know how does the approach works at different dilutions (1:2, 1:5, 1:20, 1:100...?). The authors should perform such kind of experiments.

Authors: This is indeed a good question and idea. According to your suggestion, we re-detected the detection line of *S. aureus* in artificial simulated samples, and the final concentration of bacteria was set as 50 CFU/mL, 25 CFU/mL, 10 CFU/mL, 5 CFU/mL, 2 CFU/mL, 1 CFU/mL. The results showed that no positive results could be detected below 10 CFU/mL. Which indicated that the lowest detection line of IMBS-RPA method that established in this study is 10 CFU/mL, which is the same order of magnitude as the results presented in the manuscript (2.5×10^1 CFU/mL).

⇒ **Please add this in the corresponding Mat and Methods section. I do not remember that it was added.**

Authors: Thanks for your reminding, this part has been added in the revised manuscript.

- L153: please explain why a phenol chloroform extraction is needed here

Author : RPA reaction system contains three enzymes, recombinases that bind single-stranded nucleic acids (oligonucleotide primers), single-stranded DNA-binding proteins (SSB), and strand-displaced DNA polymerases, these enzymes can affect the progress of DNA in polyacrylamide gel electrophoresis, lead to stripe dispersion. So we need to use phenol chloroform extraction to separate DNA.

⇒ **Please add this info in the Mat and methods section**

Authors: Thanks for your reminding, this part has been added in the revised manuscript.

- L263: please remove "according to the recommendations...". This is mat and methods information, and not expected in discussion!

⇒ **This modification has not been done. Please correct!**

Authors: Thanks for your reminding, the modification has been done.

- L268: reaction times

Authors: Thanks for your reminding, it has been revised

⇒ **This modification has not been done. Please correct!**

Authors: Thanks for your reminding, the modification has been done.

Line 68 reviewer 3 asked about the meaning of sample matrix.

⇒ **Please add this information in the manuscript.**

Authors: Thanks for your reminding, this part has been added in the revised manuscript.

Other minor points:

- First page: "Author order was determined on the basis of contribution".

⇒ **This is obvious and should always be the case. This sentence is not required here.**

- L54: was based **on** microbial culturing

- L55: **id**entification

- L57: were commonly used => are commonly used

- L58: **are** also used.

- L163: **were** used as specific primers for PCR.

- L168: prodedure. **Briefly** (please make a new sentence)

- L170: to Gram straining **and** coagulase test

- L206: CFU/**mL**. **When** there were only... (please make a new sentence)

- L209: CFU/**mL**. **The** capture efficiency... (please make a new sentence)

- L211: Interestingly,

- L211-212: for **a** mixed sample with the bacteria...

- L212-213: this is not correct English, please rephrase

- L236: remove "that"

- L237: shorten => shortened

- L248: by ELISA **in** spiked milk sample **that was previously** centrifuged...

- L249: CFU/**mL** **without centrifugation**.

- L251: compared with PCR, but samples needed to **be** inoculated onto...

Authors: Thanks for your reminding, these questions have been revised.

December 5, 2022

Dr. zhen wang
Beijing University of Agriculture
beijing
China

Re: Spectrum02249-22R2 (Rapid detection of *Staphylococcus aureus* in milk and pork via immunomagnetic separation and recombinase polymerase amplification)

Dear Dr. zhen wang:

Your manuscript has been accepted, and I am forwarding it to the ASM Journals Department for publication. You will be notified when your proofs are ready to be viewed.

Sincerely,

Vincenzina Fusco
Editor, Microbiology Spectrum
